# Effectiveness and Safety of a Mixture of Nobiletin and Tangeretin in Nocturia Patients: A Randomized, Placebo-Controlled, Double-Blind, Crossover Study

**DOI:** 10.3390/jcm12082757

**Published:** 2023-04-07

**Authors:** Haruki Ito, Hiromitsu Negoro, Jin Kono, Naoki Hayata, Takayoshi Miura, Yumi Manabe, Yu Miyazaki, Mutsuki Mishina, Je Tae Woo, Naoki Sakane, Hiroshi Okuno

**Affiliations:** 1Department of Urology, National Hospital Organization, Kyoto Medical Center, Kyoto 612-8555, Japan; 2Department of Urology, Institute of Medicine, University of Tsukuba, Tsukuba 305-8575, Japan; 3Department of Urology, Graduate School of Medicine, Kyoto University, Kyoto 606-8507, Japan; 4Department of Biological Chemistry, College of Bioscience and Biotechnology, Chubu University, Kasugai 487-8501, Japan; 5Division of Preventive Medicine, Clinical Research Institute, National Hospital Organization, Kyoto Medical Center, Kyoto 612-8555, Japan

**Keywords:** clinical trial, flavonoid, voiding diary

## Abstract

Nobiletin and tangeretin (NoT) are flavonoids derived from the peel of *Citrus depressa*, and they have been found to modulate circadian rhythms. Because nocturia can be considered a circadian rhythm disorder, we investigated the efficacy of NoT for treating nocturia. A randomized, placebo-controlled, double-blind, crossover study was conducted. The trial was registered with the Japan Registry of Clinical Trials (jRCTs051180071). Nocturia patients aged ≥50 years who presented nocturia more than 2 times on a frequency–volume chart were recruited. Participants received NoT or a placebo (50 mg once daily for 6 weeks), followed by a washout period of ≥2 weeks. The placebo and NoT conditions were then switched. Changes in nocturnal bladder capacity (NBC) were the primary endpoint, and changes in nighttime frequency and nocturnal polyuria index (NPi) were secondary endpoints. Forty patients (13 women) with an average age of 73.5 years were recruited for the study. Thirty-six completed the study, while four withdrew. No adverse events directly related to NoT were observed. NoT had little effect on NBC compared with the placebo. In contrast, NoT significantly changed nighttime frequency by −0.5 voids compared with the placebo (*p* = 0.040). The change in NPi from baseline to the end of NoT was significant (−2.8%, *p* = 0.048). In conclusion, NoT showed little change in NBC but resulted in decreased nighttime frequency with a tendency toward reduced NPi.

## 1. Introduction

Nocturia, defined as the number of times urine is passed during the main sleep period, is a highly bothersome lower urinary tract symptom (LUTS) [1]. The primary causes of nocturia are increased nocturnal urine volume, decreased bladder capacity, and sleep disorders, although the exact cause of each etiology is often unknown and diverse. Therefore, because of its multifactorial and complicated causes, nocturia is a disease with a high unmet medical need that is clinically difficult to cure [2]. Effective medical treatments for LUTS, such as alpha-1 blockers, beta-3 agonists, and anticholinergic agents, have been established. In contrast, only a few specific medicines for nocturia exist. For patients with nocturnal polyuria, desmopressin is effective. However, only a relatively small percentage of patients are indicated for desmopressin because there are several contraindications, such as hyponatremia, polydipsia, heart failure, renal insufficiency, and steroid/diuretic use [3]. A reliable bladder diary/frequency–volume chart (FVC) is also essential for evaluating nocturnal polyuria, which is sometimes difficult for older patients. Therefore, clinicians often find treating nocturia difficult.

Recently, there has been growing evidence that nocturia is associated with the impairment of circadian rhythm [4,5,6]. Circadian rhythms are changes in physiology and behavior that follow an approximately 24 h cycle driven by the circadian clock, and they are possessed by most living organisms, including humans. The central clock is located within the suprachiasmatic nucleus in the hypothalamus and generates the rhythm as a pacemaker. Each organ also has its own peripheral clock that modulates its circadian function [7]. From high-throughput screenings, nobiletin and tangeretin (NoT) have been identified as clock amplitude enhancers [8,9,10]. NoT are flavonoids, a kind of polyphenol obtained from the rind of *Citrus depressa*, which is a type of lemon native to Okinawa in Japan, Taiwan, and China. For example, NoT has been reported to be effective for metabolic disorders [11] or memory impairment [12] as a dietary supplement, although it is not approved for medical use in clinics. From the perspective of chronobiology, a randomized clinical trial was conducted to evaluate the effectiveness and safety of NoT in patients with nocturia (NoT-nocturia study).

## 2. Materials and Methods

### 2.1. Study Design

This randomized, placebo-controlled, double-blind, crossover study was performed at Kyoto Medical Center in Japan from November 2018 to July 2021 and included Japanese patients with nocturia. The trial was registered with the Japan Registry of Clinical Trials (jRCTs051180071), one of the primary registries in the WHO registry network, and the University Hospital Medical Information Network (UMIN000032536), one of its partner registries in it. In this study, patients who visited the outpatient clinic for nocturia and had the following inclusion criteria were recruited: age ≥50 years with nocturia confirmed by average nighttime frequency ≥2 voids on a 3-day FVC. Nocturia patients younger than 50 years would have a specific etiology other than aging; in contrast, we would like to elucidate the efficacy and safety of NoT for older adults who have multifactorial etiology along with age. Subjects were required to undergo regular medical examinations at the hospital. Patients with (1) global polyuria of 40 mL/kg/24 h or more; (2) a residual volume of 200 mL or more after voiding; and (3) an overactive bladder, severe benign prostatic hyperplasia, a neurogenic bladder, or other lower urinary tract conditions that may require its treatment, were excluded. For example, we excluded patients who were candidates for surgery or whose physicians recommended other standard treatments. Regarding patients who took any form of medicine for LUTS or nocturia, they were required to maintain this use, without any change in dose, throughout the study.

Patients were assigned to active → placebo (sequence 1, S1) or placebo → active (sequence 2, S2) sequences in a double-blind fashion by a stratified block randomization method (each block size was 4) in the pharmacy department (Figure 1). The S1 group received NoT at 50 mg orally once daily after dinner for 6 weeks, followed by a washout period of at least 2 weeks, and then the placebo orally once daily after dinner for 6 weeks; the S2 group received NoT and the placebo in reverse order. To determine NoT efficacy, we evaluated Sequence 1 in Period 1 + Sequence 2 in Period 2, and to determine placebo efficacy, we evaluated Sequence 2 in Period 1 + Sequence 1 in Period 2.

For active treatment, we used polymethoxy flavones (PMF, including 30 mg of nobiletin + 15 mg of tangeretin, and sinensetin as another ingredient). During the placebo period, patients received an identical placebo capsule containing dextrin. PMF are flavones, the phenolic hydroxyl group of which is replaced with a methoxy group. The Okinawa Research Center has established a method for manufacturing PMF90 (nobiletin, 65%; tangeretin, 25%) from the waste products of *Citrus depressa*, through separation using ethanol extraction and an alkaline treatment without agrichemicals [13]. PMF90 is now commercially available as a supplement (Nobilex^®^) in Japan.

### 2.2. Primary Endpoint Measure

The primary endpoint was a change in nocturnal bladder capacity (NBC) measured using an FVC. We calculated the mean of all voids during nighttime (including the first morning void) from a 3-day FVC.

### 2.3. Secondary Endpoint Measures

Secondary endpoints were as follows: (1) change in the International Prostate Symptom Score (IPSS); (2) sleep quality, as measured by the Pittsburgh Sleep Quality Index (PSQI); (3) changes among variables according to the FVC, nighttime frequency, the nocturnal polyuria index (NPi), and 24 h urine volume; and (4) changes in post-void residuals, blood pressure, heart rate, and salt intake. Some papers [14,15] have reported that IPSS was useful not only for men but also for women; therefore, we used IPSS for all participants. At the beginning and end of periods 1 and 2, urinary analysis was used to rule out active urinary tract infections, and levels of sodium and creatinine in spot urine specimens were measured to calculate estimated daily salt intake (g/day) according to age, height, and weight [16].

### 2.4. Statistical Analysis

Data are presented as means ± standard deviation, and statistical significance was determined using a paired *t*-test. Assuming a mean difference of 20 and a standard deviation of 20 for the change in bladder capacity between the two groups, a power level of 80% was needed to detect a significant difference at the 5% level on both sides, requiring 16 patients in each group. Taking into account dropouts, a total of 40 patients needed to be recruited. The efficacy measures for this study used a per-protocol-set analysis rather than an intention-to-treat analysis; in the per-protocol-set analysis, data were only included in the analysis if subjects followed the protocol. Statistical analysis was performed using IBM SPSS Statistics for Windows, Japanese version 27 (IBM Japan, Tokyo, Japan).

## 3. Results

### 3.1. Patient Features and Efficacy Measures

The disposition of patients and reasons for study discontinuation, according to the CONSORT statement [17], are shown in Figure 2. Among 50 patients who were initially screened, 10 were excluded, and 40 patients (27 men and 13 women) with an average age of 73.5 years were randomly assigned to the NoT/placebo (S1, *n* = 20) and placebo/NoT (S2, *n* = 20) sequences (Table 1). Among them, 36 patients from both sequences completed the second treatment period, while four patients withdrew. The reasons for withdrawal included urinary tract infection, participation in a diabetes mellitus (DM) educational program, unknown primary cancer, and poor medication compliance. Among these 36 subjects, 12 patients (33%) used anticholinergic agents or beta-3 agonists at baseline.

The mean age of the patients was 73.5 ± 5.5 years. The mean NBC, nighttime frequency, and NPi at baseline were 199.0 ± 65.6 mL, 2.78 ± 0.91 voids, and 42.5 ± 13.9%, respectively.

The primary endpoint—mean change in NBC—measured using the FVC, was 6.2 ± 38.6 mL for NoT and 0.5 ± 47.9 mL for the placebo (difference: 5.7 mL, *p* = 0.61), as shown in Figure 3. Compared with the placebo, NoT showed no significant change in the primary endpoint.

For the secondary endpoints, the variable of mean change in nighttime frequency was significantly decreased with NoT compared with that of the placebo. The mean changes were −0.49 ± 0.86 voids on NoT and 0.01 ± 0.82 voids on the placebo (difference: −0.50 voids, *p* = 0.040). Other secondary endpoints were not significantly different between NoT and the placebo. However, there was a significant change in the NPi from baseline to the end of NoT administration. The changes in the NPi were −2.8 ± 8.1% on NoT and −0.2 ± 11.6% on the placebo (difference: −2.6%, *p* = 0.30), while the before–after difference in the NPi on NoT (i.e., the active treatment period) decreased significantly from 42.7% to 39.9% (before–after difference: −2.8%, *p* = 0.048). There were also no significant changes in 24 h or nocturnal urine volume, showing a change in 24 h urine volume of −14 ± 262 mL for NoT, versus 7 ± 297 mL for the placebo (difference: −21 mL, *p* = 0.69). The change in nocturnal urine volume was −55 ± 178 mL for NoT and −0.6 ± 180 mL for the placebo (difference: −54 mL, *p* = 0.26). During the NoT active period, NPi was significantly decreased in the before–after difference (from 42.7% to 39.9%, difference: −2.8%, *p* = 0.048), as the nocturnal urine volume decreased (from 668 mL to 613 mL, before–after difference: −55 mL, *p* = 0.08) compared with the change in 24 h urine volume described above. The changes in maximum voided volume through 24 h were evaluated, and the levels were 8.3 ± 67 mL for NoT and −2.4 ± 59 mL for the placebo (difference: 10.7 mL, *p* = 0.49), with no statistical difference between the two groups. Considering the above FVC parameters, the nocturia index (Ni) and the nocturnal bladder capacity index (NBCi) were calculated. The change in Ni was −0.25 ± 0.64 for NoT and −0.01 ± 0.58 for the placebo (difference: −0.25, *p* = 0.13). The change in NBCi was −0.24 ± 0.56 for NoT and 0.01 ± 0.62 for the placebo (difference: −0.25, *p* = 0.10). Although there were no statistically significant differences in Ni and NBCi, the changes in Ni and NBCi from baseline to the end of NoT were significant: NBCi significantly decreased from 2.22 to 1.97 (before–after difference: −0.25, *p* = 0.029) and NBCi significantly decreased from 1.57 to 1.33 (before–after difference: −0.24, *p* = 0.016). The changes in hours of undisturbed sleep were 0.21 ± 0.99 h on NoT compared with 0.04 ± 0.87 h on the placebo, although the difference did not reach statistical significance (difference: 0.17 h, *p* = 0.52).

None of the evaluated IPSS parameters, including item 7 (nocturia) and QOL, were significantly different (Table 2). NoT did not significantly affect sleep outcomes as evaluated using the PSQI.

We investigated the background of good responders to NoT for nocturia. A responder was defined as a patient who showed a decrease of 0.5 or more episodes per night from baseline to the end of NoT administration. There were 17 responders (47.2%) and 19 nonresponders (52.8%). Responders had significantly lower NBC compared with nonresponders (178.3 ± 58.1 mL vs. 223.9 ± 69.2 mL, difference: −45.6 mL, *p* = 0.046). Moreover, the serum creatinine of responders was significantly lower than that of nonresponders (0.80 ± 0.16 mg/dL vs. 0.92 ± 0.17 mg/dL, difference: −0.12 mg/dL, *p* = 0.049). In fact, considering the difference from baseline to the end of NoT administration, responders had larger reductions in the NPi compared with nonresponders (−6.0 ± 8.0% vs. 0.04 ± 7.1%, difference: −6.04%, *p* = 0.026).

### 3.2. Safety

No serious adverse events related to NoT were observed or reported by participants. In this study, there were no significant effects on blood pressure or heart rate, and there were no clinically abnormal findings in the blood hematological or biochemical indices. One participant was found to have a urinary tract infection after the post-drug washout phase for NoT. He was diagnosed with vesicoureteral reflex one year after that event. One man, who was the oldest participant in this study at 84 years old, experienced general fatigue and complained of black stool. He was found to have been taking the placebo. He was later diagnosed with cancer of unknown primary origin. No adverse events appeared to have any clear link to the NoT therapy. Treatment with NoT was considered well tolerated, overall, for older adults in this study.

## 4. Discussion

We investigated the efficacy of NoT supplementation in patients with nocturia by conducting a randomized, placebo-controlled, double-blind, crossover study. NoT had little influence on NBC. However, the nighttime frequency was significantly decreased. In general, fluid intake was positively correlated with 24 h urine output, and since there was no significant change in this study (difference: −21 mL, *p* = 0.69), it is likely that fluid intake did not differ either. NoT tended to decrease NPi, which may have contributed to improvement in nocturia. The decrease in Ni and NBCi during NoT administration might indicate that the use of NoT has a subtle cooperative effect on bladder capacity with nocturnal urine volume. Many anticholinergic drugs were shown to reduce nocturia episodes by 0.1–0.2 per 24 h compared with the placebo [18], and desmopressin was shown to reduce nocturia episodes by 0.2–0.5 per 24 h compared with the placebo [19]. Therefore, the effects of NoT, which reduced nocturia episodes by 0.5 compared with the placebo, are relatively promising. No severe adverse events were observed, even in older participants. To our knowledge, this is the first study to demonstrate the efficacy of NoT administration in older adults with nocturia (with a mean age of 73.5 years), and the first randomized, placebo-controlled, double-blind study of NoT in nocturia. In clinical trials of lower urinary tract symptoms, it is well-known that the effects of placebos are not negligible [20]. Therefore, a placebo-controlled trial was required. While this study has a small sample size, we believe that the randomized design provides a high degree of evidence.

In recent years, natural extracts, such as flavonoids, have been attracting attention for their health benefits. Flavonoids are natural polyphenols that are widely found in plants [21], and NoT have many biological activities and efficacy against various diseases [10,22]. Other than LUTS, these include metabolic diseases (DM, obesity), inflammatory diseases, dermatological disorders (especially antiaging and allergies), muscle atrophy, heart failure, liver dysfunction, and dementia [12]. Because NoT have been found to increase the amplitude of oscillation in the expression of clock genes, and dysfunctions of circadian rhythm are associated with such diseases [23], NoT may confer potent protection against these diseases by modulating the circadian rhythm.

Nocturia has a multifactorial pathophysiology, although it can also be considered the degeneration of the circadian rhythm of bladder capacity, urine output, and sleep–awake cycles. In this respect, we surmised that the beneficial effects of NoT could result in decreasing nighttime voiding frequency by increasing bladder capacity, decreasing urine volume at night, and/or improving sleep quality. In fact, daytime frequency did not differ significantly (difference: +0.02 voids, *p* = 0.96), while, in contrast, nighttime frequency was significantly decreased when compared with the placebo (difference: −0.50 voids, *p* = 0.040). The improvement in nighttime frequency may have been because of the decrease in NPi, although the mechanism of action was unknown in this study. Urine production is regulated by various signals (including antidiuretic hormones, natriuretic peptides, blood pressure, and others), and the kidneys also have their own circadian clock, which may be associated with the day–night rhythm of urine production [24]. Future studies are needed to determine the mechanism of action of NoT on the rhythm of urine production.

Compared with the placebo, we showed a significant effect of NoT on nighttime frequency without any significant adverse effects. However, the efficacy was relatively small. A possible reason is that the amount of nobiletin was relatively small for the expected effects. A previous article reported significant effects of nobiletin using a dose that was 17 times higher than that used in the present study (30 mg nobiletin + 15 mg tangeretin) [11]. The dose of NoT used in our study was selected because the study was considered to include older patients, who were potentially at higher risk of adverse events compared with younger participants, as in other clinical trials [25]. To achieve a higher blood concentration of NoT during the nighttime, we chose to administer NoT in the evening. Nobiletin is absorbed soon after intake, and the peak blood concentration is reached approximately 1 h after oral administration in rats [26]. We aimed to only include nighttime as a therapeutic period and then selected 30 mg of nobiletin, which was a half-dose of 60 mg administered for healthy volunteers aged an average of 49 years in the article [13]. We adopted 30 mg of nobiletin, a higher dose than 10 mg of nobiletin, which was administered for memory impairment [12]. In another article, 2.91 mg of nobiletin (including α-linolenic acid as another ingredient) was used for participants aged 68.7 years [27]. The effect of nobiletin was seen after 6 weeks of administration in the previous report [13], so that was the time period used. In addition, many studies of lower urinary tract symptoms have shown significant efficacy at 6 weeks [28,29]. However, since there were no significant differences in the main efficacy item in the present study, it could be possible that the effect may be better recognized at 12 weeks, as reported by Yamada S. et al. [12]. In this study, no adverse events directly related to NoT were observed, even in older adults. Therefore, higher doses and longer study periods will be acceptable in future studies. There is a keen need for safer medicines/nutrients with a higher safety profile, such as NoT, to treat nocturia.

Because the causes of nocturia are multifactorial and the effects of NoT vary, it is possible that some populations may derive benefits from NoT that others may not. We evaluated the features of responders to NoT for nocturia and found that smaller bladder capacity and lower serum creatinine were significantly associated. The possible explanation for this is that patients with lower NBC at baseline were more susceptible to changes in the decrease of nighttime urine production, and NoT may function better on kidneys with adequate reserve functioning.

Although prospective and placebo-controlled, the present study had some limitations. First, this was a small, single-institution study. We assumed a mean bladder volume change difference of 20 mL between NoT and the placebo; however, unfortunately, the difference only reached 5.7 mL. Increasing the number of patients could be a way to achieve more homogeneous data and statistical differences. Second, we could not confirm which components, nobiletin or tangeretin (or both), were effective or whether they had a synergistic effect. Third, owing to the purification process, the extracts included minor contents of the peel other than nobiletin and tangeretin. However, this is an advantage from both ecological and economical perspectives because NoT can be extracted from *Citrus depressa* peel, which is a waste product.

## 5. Conclusions

NoT had little effect on NBC compared with the placebo in patients with nocturia. On the other hand, NoT showed a significant decrease in nighttime frequency and a trend toward a decrease in NPi. Because NoT have been found to be safe, even in older adults, and is a commercially available supplement, the improvement noted in this pilot study may promote further, larger clinical trials with higher doses and longer evaluation periods. Our study provides potential evidence of a novel and safe therapeutic option for nocturia.

## Figures and Tables

**Figure 1 jcm-12-02757-f001:**
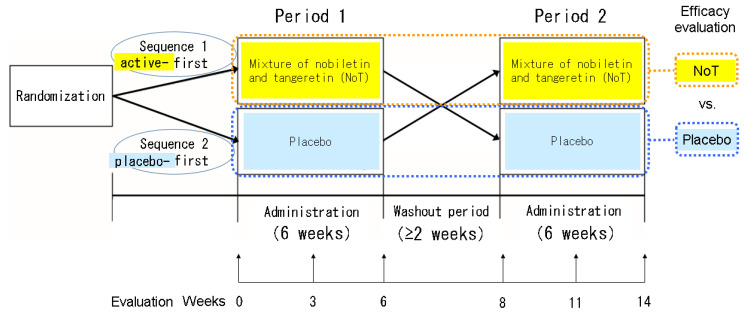
Study design.

**Figure 2 jcm-12-02757-f002:**
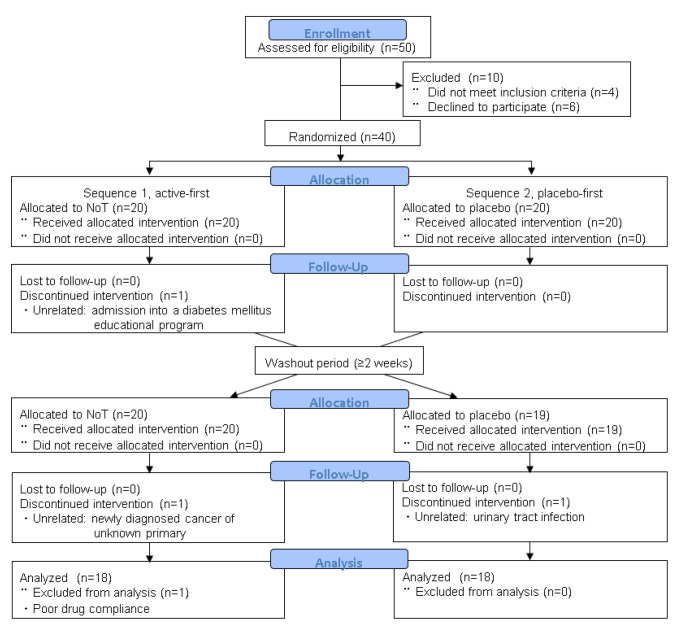
CONSORT flow diagram. No adverse events appeared to have any clear link to NoT therapy. Abbreviations: NoT, nobiletin and tangeretin.

**Figure 3 jcm-12-02757-f003:**
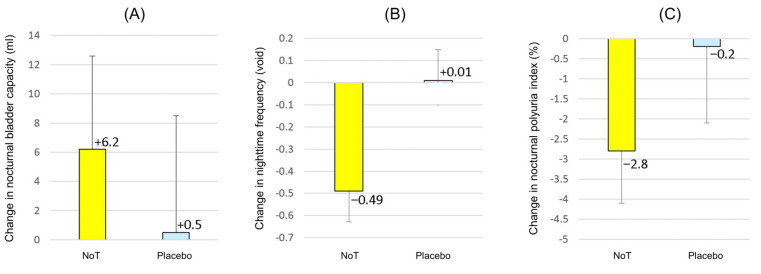
Changes in the frequency–volume chart variables from baseline. Data are presented as means ± standard error of the mean. (**A**) Nocturnal bladder capacity, (**B**) nighttime frequency, (**C**) nocturnal polyuria index. Abbreviations: NoT, nobiletin and tangeretin.

**Table 1 jcm-12-02757-t001:** Baseline features.

	Sequence 1, Active-First	Sequence 2, Placebo-First	Overall Cohort
	*n* = 20	*n* = 20	*n* = 40
Age (y)	73.8 ± 4.0	73.2 ± 6.7	73.5 ± 5.5
Sex	13 men, 7 women	14 men, 6 women	27 men, 13 women
Body mass index (kg/m^2^)	23.4 ± 3.7	24.0 ± 3.6	23.7 ± 3.6
Hypertension	13 (65%)	17 (85%)	30 (75%)
Diabetes mellitus	4 (20%)	2 (10%)	6 (15%)
Systolic blood pressure (mmHg)	133.4 ± 19.1	134.9 ± 16.6	134.2 ± 17.9
Diastolic blood pressure (mmHg)	78.5 ± 12.1	79.9 ± 7.9	79.2 ± 10.2
Heart rate (beats/min)	74.9 ± 12.1	74.9 ± 12.0	74.9 ± 12.0
Questionnaire scores			
IPSS-total	13.3 ± 6.8	10.1 ± 5.2	11.7 ± 6.3
IPSS-Q7	3.05 ± 0.59	2.55 ± 1.07	2.80 ± 0.90
IPSS-QOL	4.5 ± 1.1	4.0 ± 1.2	4.2 ± 1.1
PSQI	8.6 ± 2.9	8.8 ± 4.5	8.7 ± 3.8
Frequency–volume chart variables			
Nocturnal bladder capacity (mL)	192.5 ± 66.1	205.5 ± 64.4	199.0 ± 65.6
Nighttime frequency (void)	2.85 ± 0.72	2.72 ± 1.06	2.78 ± 0.91
24 h urine volume (mL)	1483 ± 341	1647 ± 348	1565 ± 354
Nocturnal polyuria index (%)	43.9 ± 15.2	41.0 ± 12.4	42.5 ± 13.9
Nocturnal urine volume (mL)	658 ± 298	673 ± 244	666 ± 272
Maximum voided volume (mL)	307.5 ± 87.1	301.5 ± 76.0	304.5 ± 81.8
Nocturia index *	2.19 ± 0.81	2.30 ± 0.86	2.24 ± 0.84
Nocturnal bladder capacity index ^†^	1.66 ± 0.54	1.42 ± 0.56	1.54 ± 0.57
Hours of undisturbed sleep (h)	2.31 ± 0.71	2.40 ± 0.70	2.35 ± 0.71
Laboratory data			
Post-void residual (mL)	19.8 ± 25.5	27.5 ± 31.2	23.6 ± 28.8
Serum creatinine (mg/dL)	0.90 ± 0.23	0.89 ± 0.20	0.89 ± 0.21
BNP (pg/mL)	40.5 ± 33.9	35.6 ± 33.5	38.1 ± 33.8
Estimated daily salt intake (g/day)	8.79 ± 2.38	8.77 ± 2.99	8.78 ± 2.70

* Nocturia index = nocturnal urine volume/maximum voided volume. ^†^ Nocturnal bladder capacity index = nighttime frequency − nocturia index + 1. Abbreviations: IPSS, International Prostate Symptom Score; PSQI, Pittsburgh Sleep Quality Index; BNP, brain natriuretic peptide.

**Table 2 jcm-12-02757-t002:** Changes in questionnaire scores from baseline.

	Change from Baseline			
Questionnaire Scores	NoT	Placebo	Difference	*p*-Value
	*n* = 36	*n* = 36		
IPSS-total	−0.64 ± 3.20	−0.68 ± 2.99	0.04	0.88
IPSS-Q7	−0.22 ± 0.75	−0.21 ± 0.80	−0.01	0.89
IPSS-QOL	−0.19 ± 0.99	−0.06 ± 1.11	−0.13	0.49
PSQI	−0.5 ± 1.9	−0.2 ± 2.0	−0.3	0.40

Abbreviations: NoT, nobiletin and tangeretin; IPSS, International Prostate Symptom Score; PSQI, Pittsburgh Sleep Quality Index.

## Data Availability

Not applicable.

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
