# Peer review of "Effectiveness and Safety of a Mixture of Nobiletin and Tangeretin in Nocturia Patients: A Randomized, Placebo-Controlled, Double-Blind, Crossover Study"

_jcm, 2023, doi:10.3390/jcm12082757_

Round 1
Reviewer 1 Report
This study provides novel insight into the efficacy and safety of nobiletin and tangeretin (NoT) in elder patients with nocturia. This is an interesting topic focusing on circadian dysregulation as a potential mechanism of nocturia. Overall, the study is well-designed and logically written, even with mostly negative findings at the end of the trial. There are several comments and suggestions for authors to consider.
1. Have NoT been used to treat diseases other than nocturia in the clinic? Please briefly mention it using one or two sentences with references in the introduction.
2. In your design, why choose patients above 50 instead of recruiting some younger patients. This should be discussed in detail because elder patients are more likely to suffer from urological complications and systemic comorbidities, which cause affect the treatment. I think this could be one of the main reasons to get negative findings in this trial.
3. The authors exclude patients with “severe” BPH, overactive bladder, neurogenic bladder etc. Please define “severe” in detail. Any criteria?
4. Did the authors assessed urine to rule out active infection? Have routinely urine tests done during the experiment? What did the authors do if you find out a patient had urinary infection during the course of experiment? Please specify in the study design.
5. Were those on other medications, which could possibly treat or cause nocturia excluded? If not, how long should they stop taking those drugs before participation? Please specify in the study design.
6. How did the authors determine the dosage for NoT? Any evidence on the dosage and treatment duration? Please add references to justify their use.
7. Since both men and women were recruited, the authors need to consider if using IPSS for both genders is a proper design.
8. Did the authors use the intention-to-treat analysis in this trial?
9. In the results, it might need to consider the possibility that the decrease in nighttime frequency is due to the change of fluid intake, since the bladder capacity and nocturnal polyuria index remain unchanged. However, it is difficult to explain that because of no data on water intake was recorded.
10. The relatively small sample size could be another reason to get negative results of this trial. As can be seem the long error bars in figure 3 A and C. Increasing patient number could be a way to achieve more homogeneous data and statistical differences.
Reviewer 2 Report
We congratulate you for producing very good research results. Based on the results of the above study, it is thought to be very helpful in selecting a treatment for refractory nocturia. I would like to ask you a few questions regarding the above research.
#1. Why was the sample size set to 50 when designing this prospective study? When designing a prospective study, the researcher should explain the reasons for obtaining the appropriate sample size for the study.
#2. The primary endpoint of this study is the change in NBC. The researchers concluded that there was no difference in NBC between the two groups. However, if the sample size is appropriate, I think there will be a difference in NBC between the two groups. Especially in the case of this drug, given that there was a difference in NBC between the group with and without a response, it is expected that there will be a difference in NBC between the two groups. It is thought that it would be good to reconsider the conclusions reached by the researcher. (It is also recommended to check for a difference in NBCi.)
#3. NPi is different between the two groups. But researchers couldn't explain why. NPi=NUV/MVV. Is there no difference in NUV between the two groups? Is there no difference in MVV between the two groups? You can figure out why the NPi makes a difference by checking this out. Given that NoT's efficacy improves circadian rhythm and also increases NBC, it is likely that the cause of the decreased NPi is an increase in MVV. As with nocturnal enuresis, in the case of nocturia with increased NBCi, increased NBC may increase MVV. (MVV increases when nocturnal enuresis improves in patients with nocturnal enuresis. Supplementary table 1 at SC Kim, Investig Clin Urol. 2021;62(3) 317-323) If NBC was increased using NoT, it is possible that the MVV of the patient group increased and NPi decreased. Please check the difference between NUV and MVV between the two groups.
#4. The above study is a cross-over study. In the material and method, it was explained by dividing into two groups, S1 and S2, but in the result, it is divided into the active group and the placebo group, which can cause confusion. To show the research results more effectively, serially showing the changes of NBC and NPi in active and placebo in S1 and S2 is thought to be a more effective way to explain without confusion.
